



# An assessment of domestic rice distribution for transboundary water-food management in Japan through virtual water trade

Sang-Hyun Lee[1], Makoto Taniguchi[1*], Naoki Masuhara[1], Yun-Gyeong Oh[2]

[1] Research Institute for Humanity and Nature (RIHN), Motoyama 457-4, Kamigamo, Kita-ku, Kyoto, Japan

[2] Institute of Agricultural Science & Technology, Chonnam National University, Gwangju, Republic of Korea

*Correspondence to*: Makoto Taniguchi (makoto@chikyu.ac.jp)

**Abstract**

The self-sufficiency ratio of rice in Tokyo was less than 5% but the national self-sufficiency ratio in Japan has reached over 90 % in 2012. Accordingly, local rice distribution is a key factor for regional food security but also water resource management because a significant amount of irrigation water was simultaneously used or saved in the production or consumption areas, respectively, through rice distribution. This study's aim is to analyze domestic rice distribution, and assess the dependency of prefectures on water resources within or outside the region through virtual water flows that indicate the embedded water in food trade. The results show that significant amounts of water were transferred virtually from the Northern regions such as Tohoku and Hokkaido to industrial and urbanized regions including Tokyo and Osaka through domestic rice distribution. In particular, the outflow of virtual water from the Tohoku region was more than 3 billion m³/yr during 2000–2012, and it was almost three times larger than the net flows of virtual water within the Tohoku region. In addition, increasing self-supply in the main producing prefectures affected the changes in the entire rice distribution network. In particular, as the self-supply in Shiga—the main producing prefecture sharing the Yodo-river basin with Osaka in the Kansai region—increases, the dependency of Osaka on virtual water flows from outside the Kansai region rises. Thus, it is important to consider the link between food and water through virtual water flow in trans-boundary water management, integrating not only water resource boundaries such as river basins, but also food boundaries covering production and consumption areas.

Keywords: *Paddy rice; Domestic distribution; Virtual water trade; Food self-sufficiency; Japan*

## 1 Introduction

According to the FAO, rice consumption in Japan was 112.8 kg/capita/yr in 1961 and it decreased continuously to 58.7 kg/capita/yr in 2012. Although rice consumption has decreased, the average daily calorie intake of a Japanese person in 2012 was approximately 1,870 kcal/capita/day and approximately 30% was gained from rice. In addition, about 7.1 million tons/yr of rice was consumed as food in 2012 and more than 90% of the total consumption was fulfilled by domestic production (FAOSTAT, 2018). Therefore, rice production is regarded as the main factor for food security in Japan and rice self-sufficiency has not dropped below 80 % since 1961 (MAFF, 2016). However, in terms of the regional balance of rice production and consumption, rice has been produced intensively in prefectures in the northern area such as Hokkaido, Akita, Aomori, Niigata, and Iwate, but other prefectures including mega cities such as Tokyo and Osaka have consumed significant amounts of rice with very low self-sufficiency (MLIT, 2018). Accordingly, we can infer that a significant amount of rice has been distributed among prefectures, which could affect food security and resource requirements in rice-cultivating prefectures. However, there is no available data regarding the domestic distribution of rice in Japan. Thus, it is important to estimate domestic rice distribution among prefectures through a reasonable approach such as the gravity model (Tinbergen, 1962; Pöyhönen, 1963; Rodrigue et al., 2009). The gravity model is a representative empirical method for estimating bilateral trade flows between a





pair of countries considering spatial interaction such as economic mass and distance between them (Batra, 2006). The equation
in the gravity model mimics gravitational interaction as described in Isaac Newton's law of gravity, and this model provides
an estimate of the volume of flows of, for example, goods, services, or people between two or more locations. Since this model
was introduced to analyze international trade flows by Tinbergen (1962) and Pöyhönen (1963), it has been successfully applied
to flows of varying types such as migration, foreign direct investment, and more specifically to international trade flows
(Martinez-Zarzoso, 2003). In addition, there are a huge number of empirical applications in the literature on international trade
which have contributed to the improvement of the performance of the gravity equation (Wei, 1996; Bougheas et al., 1999;
Egger, 2000; Martinez-Zarzoso, 2003; Antonucci and Manzocchi, 2006). The key strength of the gravity model is in
considering spatial interaction between traders. Spatial interactions cover a wide variety of movements, such as trips to work,
migrations, public facility usage, the transmission of information or capital, retail marketing activities, international trade, and
freight distribution (Rodrigue, Comtois, and Slack, 2009). The basic assumption behind many spatial interaction models is
that flows are a function of the attributes of the origins and destinations and the friction of the distance between them (Rodrigue
et al., 2009). In this study, we applied the spatial interaction based on distance between prefectures and production and
consumption of rice to the gravity model, and estimated the domestic rice distribution among prefectures.
Rice distribution could also be to related water management because most rice in Japan is cultivated under full irrigation in
paddy fields (MLIT, 2018), and the irrigation water may be considered as the water contribution to supplying rice to other
prefectures, and prefectures importing rice instead of producing could save irrigation water. To sum up, rice distribution could
be bound up with water supply in exporting prefectures and water saving in importing prefectures; thus, it is important to bring
rice distribution into trans-boundary water management linking exporting and importing prefectures. However, water is hardly
tradable, making it difficult to integrate prefectures in terms of water resource management. Some integrated water
management focuses on various water resources within a water basin but the trans-boundary water management referred to in
this study is more related to interlinkages among prefectures through food trade. Therefore, we applied the concepts of the
water footprint (WF) and virtual water trade (VWT) to assessing the effects of food distribution on regional water management.
The WF is the volume of water needed to produce one ton of product in the region where the product is actually produced
(Chapagain and Hoekstra, 2004). The concept, indicating the water embedded in the trade of products, was developed and
introduced to an international audience at an expert meeting on VWT in 2002, which contributed to solving water shortages
in the Middle East (Allan, 2001; Hoekstra 2003). The VWT has been used for quantifying how much water was traded between
countries in a virtual form through food trade, and it has been suggested as a relevant tool for a national water policy (Schyns
and Hoekstra, 2014). In the past, national water strategies have considered the quantification of actual water use and the
allocation of water resources, such as surface or ground water, in the context of their neighboring countries. However, VWT
suggests a wider perspective by expanding the watershed boundary from a real watershed to a virtual area between the importer
and exporters; for example, the virtual water export from a nation is the sum of the virtual water export from domestic water
resources (Fader et al., 2011; Hanasaki et al., 2010; Lee et al, 2017). In various studies, the concept of VWT has been adapted
for assessing water security considering the effects of food trade. For example, Falkenmark and Lannerstad (2010) analyzed
the effects of future agricultural water deficits through quantifying the VWT requirement and revealed that it will be necessary
to double the VWT by 2050. Biewald et al. (2014) evaluated the trade-related value of blue water usage using VWT and found
that the international trade of food crops could save blue water worth 2.4 billion US$ globally. In cases of VWT application
on national and local scales, Bulsink et al. (2010) used the VWT to explain the positive effects of food trade on the water
resilience in Java, Indonesia, and Dalin et al. (2014) quantified the volumes of VWT between provinces in China and other
countries through agricultural trade. Lee et al. (2016) evaluated the vulnerability of VWT using degree and eigenvector
centralities that included both the volume and connectivity in VWT networks, finding that Asian countries such as Japan and
South Korea have a more vulnerable VWT structure than European countries do. In particular, the volume of virtual water
imported through grain crops into Japan was 41.9 km³/yr for 1996 through 2000 (Oki et al., 2003).



Although VWT has been adapted to various scales of water research in terms of global, national, and local water securities, it
is difficult to apply VWT to regional water resource management including different water basins or aquifers because water
is basically regarded as a non-tradable resource and specific facilities such as pipelines are required for transferring water
between water basins. However, it is obvious that water is used for producing products and water management for production
could be related to the regions where the products are consumed; thus, regional VWT could have a relationship with trans-
boundary water management including production and consumption areas. For example, Shiga, Osaka, and Kyoto share the
Yodo-river water; Shiga produces significant amounts of rice and exports it to Osaka and Kyoto. Accordingly, Shiga has used
a significant amount of irrigation water, some of which could affect water management in Osaka and Kyoto, but ultimately
irrigation water was used for supplying rice to Osaka and Kyoto. Therefore, trans-boundary water management linking
production areas to consumption areas could be considered, making the concept of VWT a useful indicator for integrating food
and water management. In particular, we focused more on trans-boundary water management on a regional scale through
internal and external VWT in each region. Additionally, the food security in production and consumption areas is related to
the domestic distribution of rice, which accompanied the changes in regional VWT. For example, Chiba is the main rice
producing area in the Kanto region and supplies rice to Tokyo and Kanagawa, which are high-demand rice consumption areas
in the region. Therefore, changes in rice self-supply in Chiba for local food security could affect the internal VWT in the Kanto
region, which is related to regional water management.
Accordingly, it is important to consider food trade in the context of trans-boundary water management, and this study estimated
domestic rice distribution among prefectures during 2000–2012 through the gravity model with various food-security scenarios.
Based on the estimations, we analyzed the internal/external VWT and assessed the dependency of prefectures including mega
cities such as Tokyo and Osaka on water resources in exporting prefectures.
**2 Materials and Methods**
**2.1 Estimation of domestic rice distribution among prefectures considering spatial interaction**
Domestic rice distribution could be considered the interaction of behaviors between producers and consumers. For example,
consumers buy products from producers providing good prices, and generally producers managing big markets can provide
good prices. In addition, the distance between consumers and producers may be a factor affecting the decision of consumers
and producers in terms of transportation cost or easy accessibility. To sum up, the distribution of products should be affected
by spatial interaction, for example, a consumer generally purchases goods from the closest biggest market.
In the case of food distribution, spatial interaction is related to food balance and distances between prefectures. These were
measured using the gravity model, which utilizes the gravitational force concept as an analogy to explain the volume of trade,
capital flows, and migration among the countries of the world (Roy and Thill, 2004). Rice food balance in Japan differs by
prefecture, as shown in Table 1. The largest amount of rice is produced in the Tohoku region located in the Northeast of Japan
including the Niigata, Akita, and Fukushima prefectures among others, and approximately 637 thousand tons/yr was produced
from 2000 to 2012, which was 34 % of the total rice production. Among the 47 prefectures, Niigata is the biggest rice growing
area, producing 479.6 thousand tons/yr between 2000 and 2012. However, only 130.1 thousand tons/yr was consumed in
Niigata; thus, even though we assumed that only people living in Niigata consumed rice produced in Niigata, about 73 % of
Niigata's total production was consumed by people living in other prefectures. The next main rice-producing prefectures were
Hokkaido and Akita producing 472.3 and 397.9 thousand tons/yr, respectively. Tokyo is the representative prefecture in the
Kanto region and consumed 762.5 thousand tons/yr despite producing only 0.6 thousand tons/yr, which was the smallest
production among all prefectures. Therefore, food security in Tokyo was very low and most rice was supplied from other
prefectures. Kanagawa is also a big rice consumer in the Kanto region, consuming 515.0 thousand tons/yr but producing only
11.8 thousand tons/yr. Therefore, these prefectures need to import large amounts of rice from other prefectures, and in





particular Chiba and Tochigi might play the role of the main producers of rice in the region, producing 245.8 and 264.7
thousand tons/yr, respectively. Accordingly, we could infer that large amounts of rice might be transported from Chiba and
Tochigi to Tokyo because Chiba and Tochigi are available to export rice, and they are close to Tokyo geographically. This
interaction is regarded as a spatial interaction in the gravity model. Based on the food balance of production and consumption,
and distances among prefectures, we analyzed the spatial interaction for the weighted rice distribution:
$$SI_{ij} = AE_i \times RI_j \times r_{ij}^{-\alpha} \qquad\qquad (1)$$
where $SI$ is the spatial interaction, $i$ and j are prefectures, $AE$ is the amount of rice available for exporting to other prefectures,
$RI$ is the rice import requirement, $r$ is distance between prefectures, and $\alpha$ is the friction parameter. The distance among
prefectures (producer and consumer) is considered a friction factor in rice distribution; thus, $\alpha$ is the friction parameter.
Actually, $\alpha$ should be estimated through observed data; however, there is no data about the distribution of rice among the
prefectures, which is why we assigned 2 to $\alpha$ for weighted distribution and 0 for non-weighted distribution in this study. As
we mentioned earlier, the distance between producer and consumer could have negative effects on distribution. Actually, real
distribution follows roads or trails but it is difficult to use real roads and trails for calculating distances because of the
complexity of roads types such as local roads, highways, and others. Accordingly, we applied the Euclidian distance and set
the point located in middle of each prefecture and calculated the straight-line distance between points using the ArcMap
program.
$AE$ and $RI$ represent the market size and purchasing power, respectively; thus, they affect the positive influence on rice
distribution. However, the export availability in main producing prefectures and import requirement in main consuming
prefectures could be changed by the rice self-supply ratio (SSR) that indicates the amount of consumed rice being produced
within the prefecture. For example, even if a prefecture produces significant amounts of rice, the availability of rice for export
could be low if a significant amount of rice is consumed within the prefecture. The choice of where to purchase is left to
consumers, but it could also be related to food security policy, for example encouraging the consumption of local food. For
example, if Kyoto prefecture sets a food security goal of increasing the consumption of rice produced in the prefecture, the
amount of rice transferred from Kyoto to other prefectures would be decreased, and also the inflow of rice into Kyoto prefecture
may also be decreased. However, it is very difficult to set the exact SSR; therefore, this study analyzed the effects of each
prefecture's rice food security on the entire domestic rice distribution through applying various SSRs as scenarios. In addition,
the SSR could have a different meaning in production and consumption areas. For example, in a production area the self-
sufficiency is determined by the amount of consumption because the amount of production is larger than consumption.
Therefore, when the SSR is set at 50 %, it means that 50 % of the total consumption is provided from rice produced within the
prefecture. However, in a consumption area where the production is smaller than consumption, a 50 % SSR indicates that 50%
of total production is consumed by people living the prefecture, as shown in Eqs (2-3).
$$AE_i = DP_i(1 - SSR_i) \qquad \text{where } DP_i < DC_i \qquad (2)$$
$$RI_j = DC_j(1 - SSR_j) \qquad \text{where } DP_j > DC_j \qquad (3)$$
where $DP$ is rice production for domestic supply, $DC$ is domestic consumption of rice, $RI_j$ is the import requirement in
prefecture (j), $AE_i$ is the available export in prefecture (i), $SSR_i$ is the self-supply ratio indicating the ratio of rice production
supplied for consumption within the prefecture to total production in prefectures under-producing rice, and $SSR_j$ is the ratio
of rice supplied from within the prefecture to total consumption in prefectures over-producing rice.
To estimate the weighted distribution by spatial interaction, first we calculated the weight value of $SI$ between specific
prefectures and applied it to $RI$ to obtain the initial export of each prefecture, as shown in Eq (4). However, there was an error
that indicated a difference between total export and the $AE$ of each prefecture; therefore, we applied the initial export between
prefectures as weighted values and calibrated the export between prefectures by multiplying the weighted values with the total
available export in each prefecture, as shown in Eq (5). Finally, we iterated these processes, as shown in Eq (6) until the errors
in Eq (7) reached less than 1 % for both $RI$ and $AE$ in every prefecture.





$$Ex_{ij} = SI_{ij} \times (\sum_i SI_{ij})^{-1} \times RI_j \qquad (4)$$
$$f^1(i,j) = Ex_{ij} \times (\sum_j Ex_{ij})^{-1} \times AE_i \qquad (5)$$
$$f^{2n}(i,j) = f^{2n-1}(i,j) \times \{\sum_i f^{2n-1}(i,j)\}^{-1} \times RI_j \qquad (n>0) \qquad (6)$$
$$f^{2n+1}(i,j) = f^{2n}(i,j) \times \{\sum_j f^{2n}(i,j)\}^{-1} \times AE_i \qquad (n>0)$$
$$err_i = \sum_j f^{2n}(i,j) - AE_i \qquad (7)$$
$$err_j = \sum_i f^{2n+1}(i,j) - RI_j$$
where $Ex_{ij}$ is the export from a prefecture ($i$) to prefecture ($j$), $SI_{ij}$ is the spatial interaction, $RI_j$ is requirement of import in a
prefecture ($j$), $AE_i$ is the available export in prefecture ($i$), $f^n(i,j)$ is the iteration function of export from a prefecture ($i$) to
prefecture ($j$), and n is the number of iterations. $err_i$ is the difference between the sum of simulated exports and $AE$ in
prefecture ($i$), and $err_j$ is the difference between the sum of simulated exports and $RI$ in prefecture ($j$).
**2.2 VWT through rice distribution**
We applied the concept of virtual water to estimate VWT through rice distribution in Japan in the context of food and water
management. First, we surveyed the regional rice WF based on water withdrawal and rice production. Water withdrawal for
paddy rice is different in each prefecture because of climate, productivity, farming technology, and so on. Ministry of land
infrastructure transport and tourism (MLIT) in Japan provides irrigation water withdrawal in rice paddy fields for the nine
regions (https://www.e-stat.go.jp), as shown in Table 1. The Kansai region, including Shiga, Kyoto, Nara, Osaka, Hyogo, and
Wakayama prefectures, withdrew 56 billion m³/yr of water in rice paddy fields from 2000 to 2012.
Water withdrawal includes not only water used in paddy fields but also return flows to rivers. In Japanese rice paddy fields,
70 % of the total water withdrawal was returned to rivers (Masumoto and Yoshida, 2014). Accordingly, 30 % of the total water
withdrawal was defined as irrigation water in this study. Figure 1 shows the regional rice WF based on production and water
withdrawal from 2000 to 2012. The minimum WF was seen in the Tohoku region (1505 m³/ton) and the maximum in Shikoku
(2253 m³/ton). Thus, cultivating rice in the Tohoku region could have a positive effect on national water security because of
the high water-use efficiency. However, we also need to consider yield as well. For example, the rice WF in the Kanto region
was 1672 m³/ton, which was smaller than other regions except for Tohoku, but the yield in the Kanto region was 5.0 tons/ha
which was the second smallest. Therefore, the Kanto region uses less irrigation water but more land is required. Tohoku
showed the smallest WF and the largest rice yield; thus, increasing rice production in Tohoku instead of other regions could
have a positive effect on national water and land savings. However, the increase of rice production in Tohoku may be a burden
to regional water security despite national water saving; hence, we analyzed the VWT among regions based on rice WF and
distribution.
The excess rice production in a prefecture is transported to other prefectures as food supply and the water used for this excess
can be regarded as a water contribution from producing prefectures to other prefectures through VWT. By contrast, importing
rice from other prefectures could have positive effects on water saving in the under-producing prefecture. As described earlier,
VWT indicates the water embedded in food trade; thus, it is calculated using the WF, as shown in Eq (8).
$$VWT_{ij} = WF_i \times Ex_{ij} \qquad (8)$$
where $VWT_{ij}$ is the amount of virtual water exported from $i$ to $j$ prefectures, $WF_i$ is the rice WF in producing prefecture $i$, and
$Ex_{ij}$ is the amount of rice exported from $i$ to $j$ prefectures.



## 3 Results and Discussion

### 3.1 Estimation of weighted rice distribution among the 47 prefectures through spatial interaction

The domestic rice distribution between the 47 prefectures in 2000–2012 was estimated based on the spatial interaction by import requirement, available export, and distance among prefectures. First, the effect of the distance on rice distribution was analyzed by applying the distance as a weight factor in spatial interaction. For example, non-weighted distribution applied the spatial interaction excluding the distance factor, whereas weighted distribution considered all variables including the distance in spatial interaction. In addition, we analyzed the effect of rice food security in all prefectures by setting various SSRs.

In the case of non-weighted distribution, the rice was transported mainly between big exporters and consumers even if they were far from each other, as shown in Figure A1 in Appendix A. For example, under a 20 % SSR scenario, the largest rice distribution is seen in Tokyo, which imported 54.47 thousand tons/yr of rice from Niigata, and the next largest was Hokkaido (50.98 thousand tons/yr). Osaka was also a big importer, and 35,4 thousand tons/yr of rice was imported from Niigata. In terms of main producing prefectures, rice production and consumption in Niigata were 479.6 thousand tons/yr and 130.1 thousand tons/yr, respectively. Therefore, 471.58 thousand tons of rice was available for exporting to other prefectures under a 20 % SSR scenario, and Niigata's excess rice production was distributed to main consumers such as Tokyo and Kanagawa in the Kanto region, Osaka and Kyoto in the Kansai region, and Aichi in the Chubu region, among others.

However, when the distance was applied as a friction factor in weighted distribution, the internal distribution within each region was increased, as shown in Figure A2 in Appendix A. For example, under a 20 % SSR scenario, the distribution from Hyogo to Osaka in Kansai region was increased from 33.13 thousand tons/yr to 55. 96 thousand tons/yr by applying weighted distribution, and the largest distribution was seen from Chiba to Tokyo, which was 99.39 thousand tons/yr, and both prefectures are located in the Kanto region. The biggest change in weighted distribution is seen in the Kyushu region. Internal distribution within the Kyushu region was smaller than other regions, and the biggest internal distribution was 56.7 thousand tons/yr from Kumamoto to Fukuoka. However, a large increase was seen in the distribution among Saga, Fukuoka, Kumamoto, and Nagasaki in the weighted distribution. For example, the amount of rice exported from Kumamoto to Fukuoka increased from .567 thousand tons/yr to 5.04 thousand tons/yr under a 20 % SSR scenario, and weighted distribution from Saga to Fukuoka also increased significantly by approximately 12 times the non-weighted distribution.

### 3.2 Analysis of impacts of food security in prefectures on the entire rice distribution through various SSR scenarios

The regional food-security policy was a main factor in rice distribution, thus we set various SSRs representing food security from 20 % to 100 % and analyzed the changes of weighted rice distribution among prefectures, as shown in Figures A2–A4 in Appendix A. As the SSR increased from 20 % to 60 %, the self-supply in each prefecture increased observably. The biggest self-supply was seen in Hokkaido (167.370 thousand tons/yr) and Chiba (147.51 thousand tons/yr). In particular Chiba was the main rice producer in the Kanto region; thus, the rice supply in Tokyo and Kanagawa, both located in the Kanto region, was strongly dependent on Chiba's rice production. Therefore, increasing the SSR in Chiba could affect the internal distribution within the Kanto region, and the amount of rice transported from Chiba to Tokyo was decreased from 99.39 thousand tons/yr to 54.28 thousand tons/yr by increasing the SSR from 20 % to 60%. Accordingly, Tokyo needs to import more rice from other regions close by, such as the Niigata and Akita prefectures in the Tohoku region. In the Kansai region, Hyogo was the main exporter to other prefectures such as Osaka and Kyoto, and when the SSR increased from 20 % to 60%, the rice self-supply in Hyogo increased from 29.86 thousand tons/yr to 89.57 thousand tons/yr, and rice transported from Hyogo to Osaka decreased from 55.96 thousand tons/yr to 34.13 thousand tons/yr.

Rice import is more significant in the main consumption areas in terms of food security; hence, we focused more on the changes in rice distribution from applying distance and SSR in Tokyo and Osaka, which are strongly dependent on rice imports. Tokyo is the biggest rice consumer and consumes 762.5 thousand tons/yr. However, only 0.6 thousand tons/yr was provided





from within Tokyo; thus, about 760 thousand tons/yr should be imported domestically. Figure 2 and Table 3 show the changes
in rice imports in Tokyo. In the non-weighted distribution with a 20 % SSR scenario, Tokyo largely imported rice from Tohoku
and Hokkaido regions which are main producers of rice in the Northern area. However, when the distance among prefectures
was applied as friction in the weighted distribution, Tokyo's imports from the Kanto region, which it belongs to, increased
significantly. For example, before applying distance as friction in the gravity model, Tokyo imported 111.7 thousand tons/yr
from the Kanto region which increased to 252.7 thousand tons/yr in the weighted distribution with a 20 % SSR scenario. As
the rice SSR increased, Tokyo's dependency on the Kanto region decreased and more rice was imported from the Tohoku
region instead. In the weighted rice distribution with a 60 % SSR scenario, 190.9 thousand tons/yr was imported from the
Kanto region and it decreased to 108.1 thousand tons/yr in a 100 % SSR scenario. We infer the two reasons for these changes:
first, Tokyo increased its self-supply but the small rice production in Tokyo had little effect. The other reason is that the main
producers in the Kanto region increased their self-supply which was accompanied by a decrease in rice exports to other
prefectures even though they are close to Tokyo. However, the imports from the Tohoku region to Tokyo were increased by
the increase in the SSR. For example, 291.3 thousand tons/yr was imported from the Tohoku region in a 20% SSR scenario
and this increased to 492.2 thousand tons/yr by increasing the SSR to 100 %. The prefectures in the Tohoku region produced
several times more than their consumption; thus, even if the SSR was increased to 100 %, they could still afford to export rice
to other prefectures, in particular to Tokyo, which is closer than other consumer prefectures.
Osaka consumed 498.7 thousand tons of rice annually between 2000 and 2012 but only produced 22.6 thousand tons. Therefore,
the rice supply in Osaka is highly related to rice imports from other prefectures. Figure 3 and Table 3 show the changes in rice
imports to Osaka. In the non-weighted distribution with a 20 % SSR scenario, 157.7 thousand tons/yr, which was 32 % of the
total consumption, would be imported from the Tohoku region in the Northern area. However, when distance was applied to
weighted distribution with a 20 % SSR scenario, importing from the Tohoku region was decreased to 82.5 thousand tons/yr
but importing from prefectures in the Kansai region that Osaka belongs to was increased from 28.5 thousand tons/yr to 144.8
thousand tons/yr. When the SSR was increased from 20 % to 60% in all prefectures, Shiga and Hyogo increased their rice self-
supply, and rice imports from Kansai to Osaka decreased to 100.7 thousand tons/yr. Therefore, Osaka needed to supplement
the amount of rice imported from the Kansai region and imports from the Tohoku region were increased to 122.4 thousand
tons/yr. When the SSR increased to 100%, this trend became more serious and more rice had to be imported from Tohoku
(206.4 thousand tons/yr).
**3.3 Assessment of rice distribution effects on trans-boundary water management through regional VWT**
*3.3.1 Analysis of internal and external VWT in regions through domestic rice distribution*
This study converted the domestic rice distribution to VWT. Regional VWT could be regarded as an important factor for trans-
boundary water management connecting rice production and consumption regions.
Figure 4 shows the VWT among regions, and the weighted distribution taking the distance factor into account derived the
increase in internal virtual water flows within regions and the decrease of virtual water flows among regions. However, changes
in the SSR caused varying situations in regional VWT. For example, as each prefecture tried to increase their self-supply rather
than rice trade by increasing their SSR, the internal VWT was increased; however, the VWT from Tohoku to the Kansai region
decreased slightly. In the Kanto region, the increase in the SSR decreased the VWT from Hokkaido, but the VWT from Tohoku
increased.
Table 4 shows more details of the internal and external VWT changes in each region for each SSR scenario. The largest internal
trade of virtual water is seen in the Kanto region. Here, the internal VWT reached 1754 Mm³ with the 20 % SSR scenario, and
it increased to 2018 Mm³ with the 100 % SSR scenario. The Kyushu region shows an internal VWT of 1387 Mm³ with a 20 %
SSR, which was the second largest, and it increased to 1697 Mm³ with a 100 % SSR. A small internal VWT is evident in
Hokkaido and Shikoku. However, the largest export of virtual water is seen in the Tohoku and Hokkaido regions. For example,





3229 Mm³ of virtual water was exported from the Tohoku region through the weighted rice distribution with an SSR 20%
scenario, and it decreased to 3052 Mm³ when all prefectures were set to a 100 % SSR food policy. The largest decrease of
virtual water export caused by an increase in the SSR is seen in Hokkaido, where it decreased from 1061 to 493 Mm³. In the
case of virtual water import, the Kanto region imported the largest amount of virtual water (2755 Mm³/yr) which was about
37 % of the total virtual water import in Japan. The next largest region was Kansai, and 1683 Mm³ of virtual water was
imported from outside Kansai through weighted rice distribution with a 20% SSR scenario.
*3.3.2 Trans-boundary water dependency in mega-prefectures such as Tokyo and Osaka through domestic rice distribution*
Tokyo and Osaka imported a significant amount of virtual water with a high dependency on water resources in other prefectures;
thus, VWT in these prefectures could strongly affect regional water management. For example, Osaka shares the Yodo-river
basin with Kyoto, Nara, and Shiga. In particular, Shiga is the main rice producer and exports rice to Osaka. Therefore, the
distribution of rice from Shiga to prefectures located in the Kansai or other regions could be important factors for trans-
boundary water management. Currently the largest portion of the population lives in Tokyo and the rice import to Tokyo could
be highly related to water resource management in prefectures located in the Kanto and other regions.
Accordingly, we analyzed the dependency on internal and external virtual water imports in Tokyo and Osaka. When the
distance was applied to rice distribution as a friction factor, the amount of rice distributed from the Kanto region to Tokyo
increased significantly. However, as the SSR increased from 20 % to 100 %, the dependency on water resources in the Kanto
region decreased and significant amounts of virtual water were transferred from the Tohoku region, as shown in Figure 5. For
example, 450 thousand m³/yr was imported from the Kanto region in a 20 % SSR scenario, and it decreased to 219 thousand
m³/yr with the 100 % SSR scenario. By contrast, 39 % of total virtual water imported to Tokyo was transferred from the
Tohoku region under the 20% SSR scenario but it increased to 66 % under the 100 % SSR scenario. Therefore, the effect of
an increase in food security in all prefectures could mean that Tokyo depended less on the Kanto region for water. In the case
of Osaka, water dependency on the Kansai region increased significantly in the weighted rice distribution with a 20 % SSR
scenario in comparison with the non-weighted scenario. For example, virtual water imported from the Kansai region to Osaka
increased from 78 thousand m³ to 369 thousand m³ by applying weighted distribution, which was 35 % of the total virtual
water imported by Osaka. Therefore, Osaka has a large influence on water management in the Kansai region and trans-
boundary water management linked to Osaka's rice imports should be considered. In addition, as the SSR was increased from
20 % to 60 %, virtual water transferred from Kansai to Osaka was decreased by 282 thousand m³/yr, and it dropped to 133
thousand m³/yr under the 100 % SSR scenario. However, virtual water imported from the Tohoku region increased to 374
thousand m³/yr, which was 2.8 times more than the virtual water imported from the Kansai region. Therefore, the dependency
of high consumption prefectures such as Tokyo and Osaka on regional water management could be affected by transportation
distance and SSRs in other prefectures. Hence, this study shows that regional water management should include not only water
use in production areas, but also the entire rice distribution network and VWT including consumption areas.
**4. Conclusions**
In Japan, the area under rice cultivation has decreased since 1960, with the area in 2016 being only 44% of that in 1960 (World
Bank, 2018), and rice is likely to be more sensitive to global warming than other agricultural crops and livestock, and over 70%
of prefectures in Japan recognize the warming effects on rice (Sugiura et al., 2012). The decrease in rice production could pose
a serious problem for food supply in some areas, thus the Japanese national government set a policy for maintaining the level
of the rice self-sufficiency ratio as close to 100% as possible (MAFF, 2015). However, the regional rice self-sufficiency ratio
in main consumption areas such as Tokyo and Osaka, was less than 5%; thus, most of their rice supply is dependent on imports
from other prefectures. This means that domestic rice distribution could have positive effects on food contribution in terms of





food security as well as water saving in consuming prefectures. By contrast, the supply of rice to other prefectures could be a
burden to water and land resource management in the main producing prefectures such as Niigata and Akita. For example, a
significant amount of irrigation water is required in rice paddy fields compared with other crops.
Therefore, this study considered rice distribution from the perspectives of trade-offs between food and water security through
the food and virtual water trade. We analyzed the changes in domestic rice distribution by spatial interaction through the
gravity model and SSR scenarios, and found that the changes in SSRs in producing and consuming prefectures could affect
the entire rice distribution as well as the internal and external virtual water flows in each region. For example, in the Kansai
region, the increase in rice self-supply in Shiga and Hyogo caused a decrease in the export of rice to Osaka, and resulted in an
increase in Osaka's rice imports from outside of the Kansai region. Accordingly, the internal virtual water flows from the
Kansai region to Osaka decreased and Osaka became less dependent on water resources in Kansai. The regional water supply
for rice production in the Kansai region is less related to rice consumption in Osaka; thus, other purposes for regional water
use might be suggested by Osaka and these should be discussed between Osaka and the main producing prefecture in the
Kansai region, such as Shiga. Therefore, the results from this study provide useful information for regional water management
and the integration of consumption and production areas.
However, there are limitations; for example, the gravity model used for estimating domestic rice distribution could not be
validated because of the lack of observed data even though we attempted to follow a reasonable approach. Second, this study
only focused on rice distribution with no changes in total production and consumption; therefore, the self-supply scenarios
applied in this study did not involve additional production but only affected the self-supply ratio in total production or
consumption. Third, the regional difference in rice consumption was not considered because of lack of data. We assumed that
people in Japan consumed the same amount of rice per year even if they lived in different prefectures.
In order to overcome these limitations, we applied the gravity model that was well-known approach for estimating products
trade, and WF of rice was calculated through the reliable statistic data from Japanese government. In addition, we tried to
consider various situations in domestic rice distribution through SSRs scenario relating to food security policy. In spite of
these limitations, the attempt to estimate domestic rice distribution and apply it to regional VWT in this study may contribute
to developing a useful indicator connecting water and food management in a framework for bridging scientists and policy-
makers such as the Water-Energy-Food Nexus, and emerging as an appropriate tool for accomplishing the sustainable
development goals.

**Data availability:** The results data for this study are freely available by contacting the corresponding author.

**Author contribution:** Sanghyun Lee, Makoto Taniguchi, and Naoki Masuhara conceived and designed the research; Sang-
Hyun Lee and Naoki Masuhara analyzed the data; Sanghyun Lee and Yun-Gyeong Oh contributed analysis tools; Sanghyun
Lee and Makoto Taniguchi wrote the paper.

**Competing interests:** The authors declare that there are no conflicts of interest regarding this publication.

**Acknowledgement:** This study was supported by the Japan Science and Technology Agency as part of the Belmont Forum
and "Research Program for Agricultural Science & Technology Development (Project No. PJ013435)".

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





**Table 1** Annual rice production and consumption as food in the 47 prefectures from 2000 to 2012

| Region | Prefecture | Consump-tion | Produc-tion | Self-sufficient (Prod./Con.) | Region | Prefecture | Consump-tion | Produc-tion | Self-sufficient (Prod./Con.) |
|---|---|---|---|---|---|---|---|---|---|
| | | 1000 tons/yr | 1000 tons/yr | % | | | 1000 tons/yr | 1000 tons/yr | % |
| Chubu | Aichi | 422.3 | 118.8 | 28% | Kanto | Chiba | 351.2 | 245.8 | 70% |
| | Fukui | 44.4 | 107.5 | 242% | | Gunma | 111.3 | 69.3 | 62% |
| | Gifu | 114.7 | 94.0 | 82% | | Ibaraki | 164.6 | 301.2 | 183% |
| | Ishikawa | 65.1 | 103.7 | 159% | | Kanagawa | 515.0 | 11.8 | 2% |
| | Mie | 102.5 | 119.3 | 116% | | Saitama | 409.7 | 135.5 | 33% |
| | Nagano | 118.5 | 165.3 | 139% | | Tochigi | 111.4 | 264.7 | 238% |
| | Shizuoka | 208.8 | 71.3 | 34% | | Tokyo | 762.5 | 0.6 | 0% |
| | Toyama | 60.2 | 161.9 | 269% | | Yamanashi | 47.1 | 22.1 | 47% |
| Chugoku | Hiroshima | 160.5 | 105.1 | 65% | Kyushu | Fukuoka | 287.9 | 151.8 | 53% |
| | Okayama | 108.5 | 135.9 | 125% | | Kagoshima | 93.0 | 91.8 | 99% |
| | Shimane | 39.2 | 76.6 | 195% | | Kumamoto | 100.8 | 158.2 | 157% |
| | Tottori | 32.4 | 55.7 | 172% | | Miyazaki | 62.3 | 76.9 | 123% |
| | Yamaguchi | 79.3 | 91.7 | 116% | | Nagasaki | 77.8 | 51.1 | 66% |
| Kansai | Hyogo | 312.4 | 149.3 | 48% | | Oita | 65.9 | 96.4 | 146% |
| | Kyoto | 147.3 | 61.9 | 42% | | Saga | 47.0 | 112.4 | 239% |
| | Nara | 77.0 | 37.1 | 48% | Tohoku | Akita | 57.7 | 397.9 | 690% |
| | Osaka | 498.7 | 22.6 | 5% | | Aomori | 73.9 | 225.6 | 305% |
| | Shiga | 79.7 | 133.3 | 167% | | Fukushima | 108.0 | 317.9 | 294% |
| | Wakayama | 54.4 | 28.9 | 53% | | Iwate | 72.2 | 233.8 | 324% |
| Shikoku | Ehime | 78.2 | 59.9 | 77% | | Miyagi | 131.7 | 302.5 | 230% |
| | Kagawa | 55.1 | 57.4 | 104% | | Niigata | 130.1 | 479.6 | 369% |
| | Kochi | 41.1 | 46.6 | 113% | | Yamagata | 63.4 | 310.8 | 490% |
| | Tokushima | 42.7 | 49.8 | 117% | Okinawa | Okinawa | 80.9 | 2.4 | 3% |
| Hokkaido | Hokkaido | 303.8 | 472.3 | 155% | Total | | 7172.1 | 6585.9 | 92% |


**Table 2** Water withdrawal in rice paddy fields

| Regions | Water withdrawal in rice paddy fields (billion m³/yr) | | | | | | |
|---|---|---|---|---|---|---|---|
| | 2000 | 2005 | 2009 | 2010 | 2011 | 2012 | Average |
| Hokkaido | 4.5 | 4.3 | 4.3 | 4.3 | 4.3 | 4.3 | 4.4 |
| Tohoku | 15.4 | 14.9 | 14.9 | 14.8 | 14.8 | 14.6 | 14.9 |
| Kanto | 8.0 | 7.7 | 7.7 | 7.6 | 7.6 | 7.6 | 7.7 |
| Chubu | 7.9 | 7.5 | 7.3 | 7.3 | 7.3 | 7.3 | 7.4 |
| Kansai | 4.1 | 3.9 | 3.9 | 3.8 | 3.9 | 3.8 | 3.9 |
| Chugoku | 4.4 | 4.0 | 4.0 | 4.0 | 4.0 | 3.9 | 4.0 |
| Shikoku | 2.4 | 2.1 | 2.1 | 2.1 | 2.1 | 2.0 | 2.1 |
| Kyushu | 7.3 | 6.9 | 6.9 | 6.9 | 6.9 | 6.9 | 6.9 |
| Okinawa | 0.1 | 0.2 | 0.2 | 0.3 | 0.3 | 0.2 | 0.2 |

Source: Ministry of Land, Infrastructure, Transport and Tourism (MLIT)







**Table 3** Import of rice in Osaka and Tokyo from regions by non- and weighted distributions with SSR scenarios during 2000–2012

| Importer (prefecture) | Exporters (region) | Non-weighted import (1000 tons/yr) SSR 20% | Weighted import (1000 tons/yr) | | | | |
|---|---|---|---|---|---|---|---|
| | | | SSR 20% | SSR 40% | SSR 60% | SSR 80% | SSR 100% |
| Osaka | Kansai | 28.5 | 144.8 | 125.0 | 100.7 | 70.4 | 32.3 |
| | Chubu | 65.5 | 99.6 | 96.9 | 93.0 | 87.1 | 77.7 |
| | Chugoku | 32.3 | 38.1 | 38.3 | 38.3 | 37.6 | 35.1 |
| | Hokkaido | 32.8 | 31.4 | 32.6 | 33.8 | 34.8 | 35.2 |
| | Kanto | 73.1 | 13.5 | 13.5 | 13.4 | 13.1 | 12.7 |
| | Kyushu | 51.4 | 20.2 | 21.0 | 21.9 | 23.1 | 24.6 |
| | Okinawa | 0.2 | 0.2 | 0.2 | 0.1 | 0.1 | 0.0 |
| | Shikoku | 14.9 | 23.1 | 22.2 | 20.7 | 17.7 | 11.3 |
| | Tohoku | 157.7 | 82.5 | 99.2 | 122.4 | 155.9 | 206.4 |
| Tokyo | Kanto | 111.7 | 252.7 | 224.1 | 190.9 | 152.5 | 108.1 |
| | Chubu | 100.1 | 72.8 | 66.9 | 59.5 | 50.3 | 38.7 |
| | Chugoku | 49.4 | 5.9 | 5.6 | 5.2 | 4.7 | 3.8 |
| | Hokkaido | 50.2 | 60.4 | 59.4 | 57.3 | 53.6 | 47.8 |
| | Kansai | 46.0 | 6.3 | 5.4 | 4.3 | 3.1 | 1.7 |
| | Kyushu | 78.5 | 7.8 | 7.6 | 7.3 | 6.9 | 6.4 |
| | Okinawa | 0.3 | 0.1 | 0.1 | 0.1 | 0.1 | 0.0 |
| | Shikoku | 22.7 | 2.8 | 2.5 | 2.2 | 1.7 | 0.9 |
| | Tohoku | 241.1 | 291.3 | 328.2 | 372.9 | 426.9 | 492.2 |


**Table 4** Internal and external virtual water flows by region through domestic rice distribution during 2000–2012

| Regions (million m³/yr) | SSRs | Chubu | Chugoku | Hokkaido | Kansai | Kanto | Kyushu | Shikoku | Tohoku |
|---|---|---|---|---|---|---|---|---|---|
| Available water resource | - | 8529 | 3282 | 5633 | 3072 | 3927 | 6214 | 2767 | 8675 |
| Net internal flow of virtual water within a region | 20% | 1017 | 485 | 142 | 798 | 1754 | 1387 | 227 | 884 |
| | 40% | 1160 | 586 | 284 | 874 | 1830 | 1461 | 295 | 943 |
| | 60% | 1307 | 686 | 427 | 941 | 1900 | 1536 | 364 | 993 |
| | 80% | 1460 | 788 | 569 | 996 | 1963 | 1612 | 436 | 1032 |
| | 100% | 1621 | 894 | 711 | 1036 | 2018 | 1691 | 514 | 1061 |
| External flow of virtual water | Outflow to other regions 20% | 1044 | 645 | 1061 | 267 | 363 | 527 | 354 | 3229 |
| | 40% | 900 | 544 | 919 | 191 | 287 | 453 | 285 | 3169 |
| | 60% | 753 | 444 | 777 | 124 | 217 | 378 | 216 | 3120 |
| | 80% | 601 | 342 | 635 | 69 | 154 | 301 | 145 | 3080 |
| | 100% | 440 | 236 | 493 | 29 | 98 | 223 | 66 | 3052 |
| | Inflow from other regions 20% | 1212 | 453 | 413 | 1683 | 2755 | 334 | 277 | 235 |
| | 40% | 1066 | 349 | 309 | 1596 | 2662 | 261 | 215 | 156 |
| | 60% | 918 | 245 | 205 | 1514 | 2572 | 189 | 153 | 90 |
| | 80% | 769 | 143 | 102 | 1435 | 2485 | 116 | 90 | 38 |
| | 100% | 615 | 40 | - | 1357 | 2402 | 45 | 23 | - |







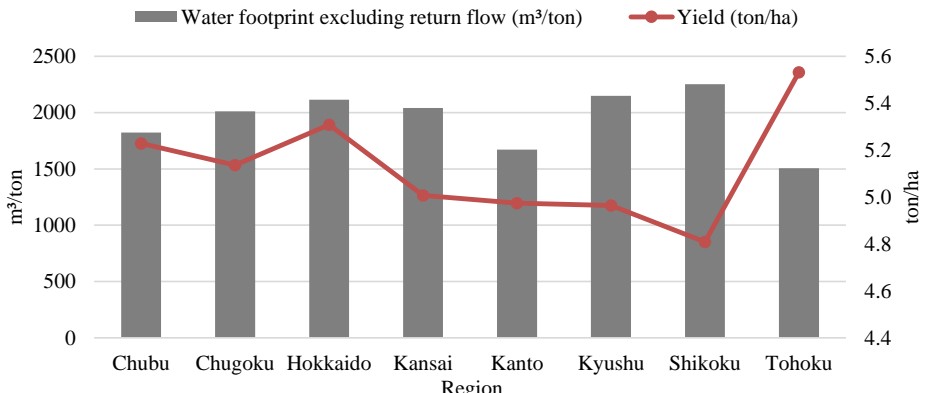


**Figure 1** Water footprint and yield of paddy rice at 8 regions in Japan










(a) Non-weighted distribution with 20 % SSR

(b) Weighted distribution with 20 % SSR

(c) Weighted distribution with 60 % SSR

(d) Weighted distribution with 100 % SSR

**Figure 2** Annual import of rice in Tokyo prefecture by non- and weighted **distribution** with SSRs during 2000–2012 (1000 **tons**/yr)





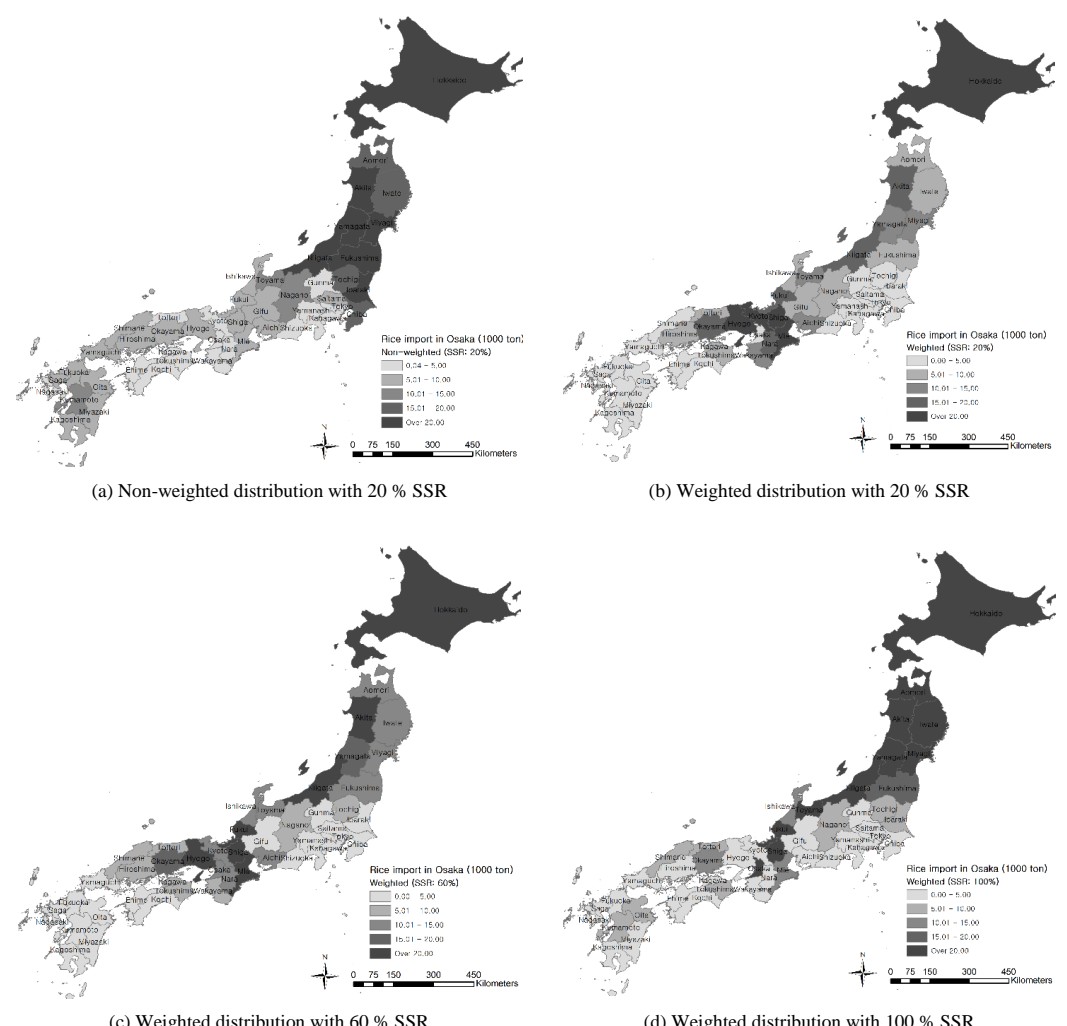

Figure 3 Annual import of rice in Osaka prefecture by non- and weighted **distribution** with SSRs during 2000–2012 (1000 **tons**/yr)



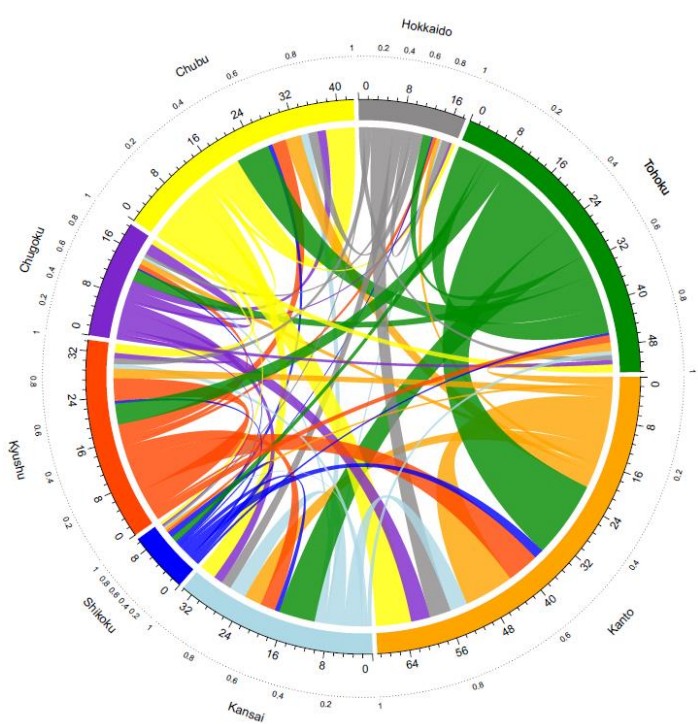

442                               (a) Non-weighted VWT (SSR: 20%)

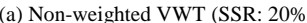

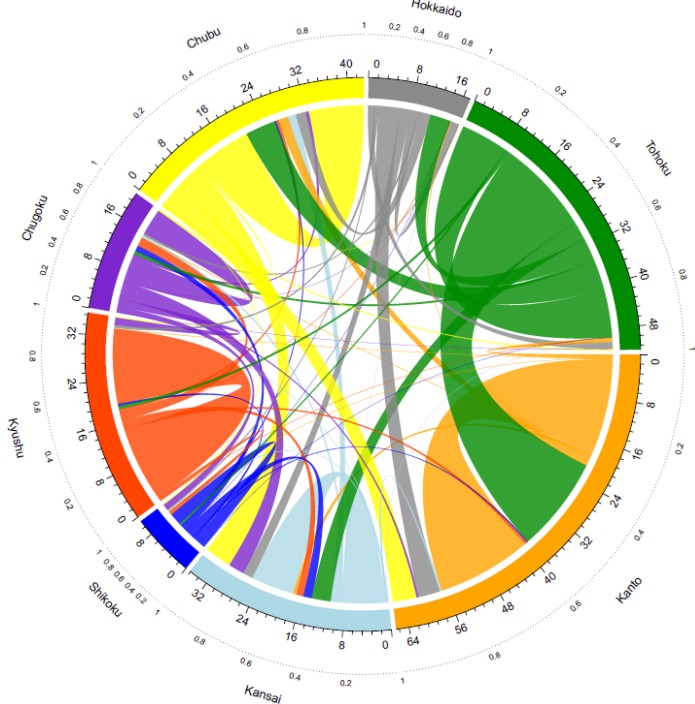

444                               (b) Weighted VWT (SSR: 20%)






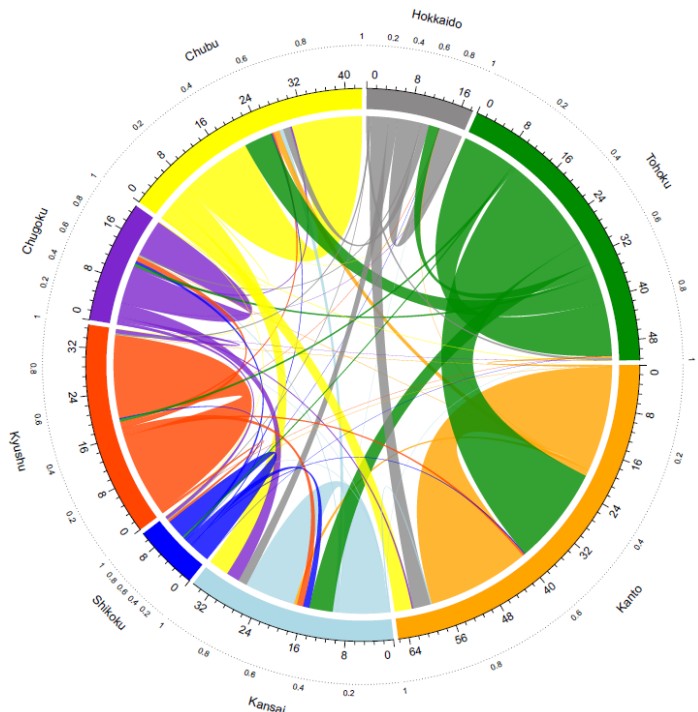


(c) Weighted VWT (SSR: 60%)

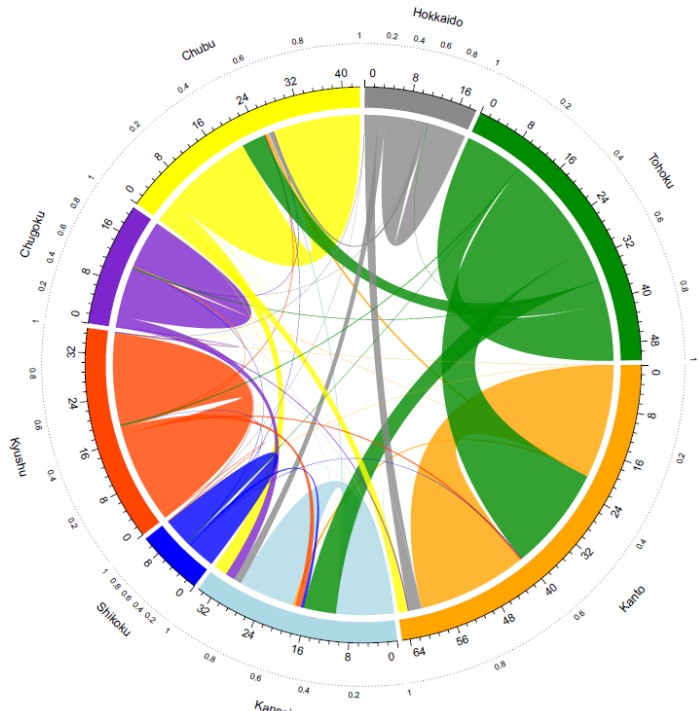


(d) Weighted VWT (SSR: 100%)

**Figure 4** Annual virtual water flow among regions through domestic rice distribution during 2000–2012. The numbers on the inner circle
indicate the amount of virtual water traded (100 million m³/yr) and the numbers on the outside circle are the percentage from 0 to 1.



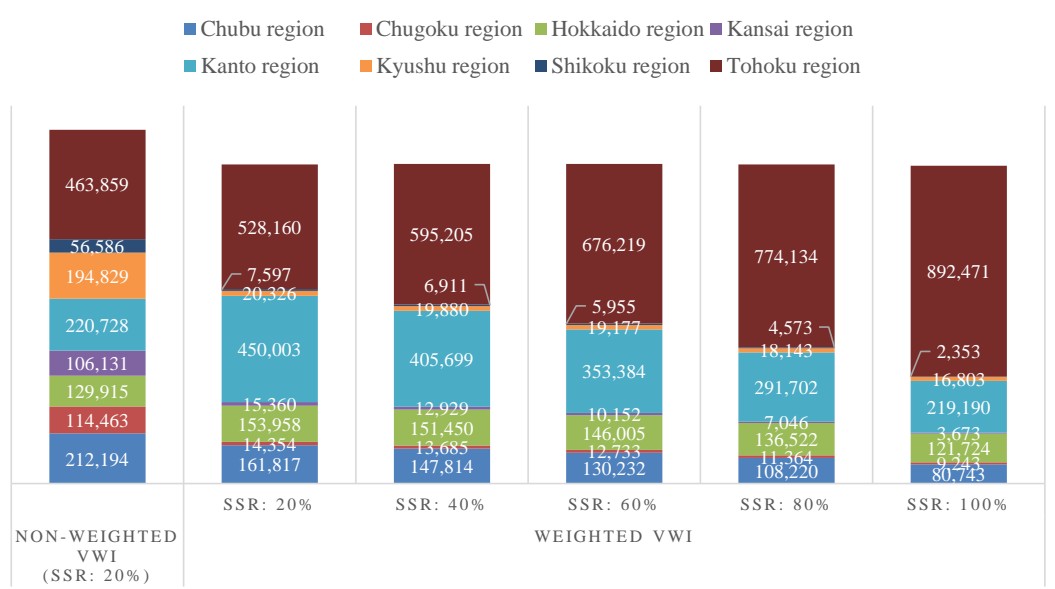

(a) Virtual water import in Tokyo

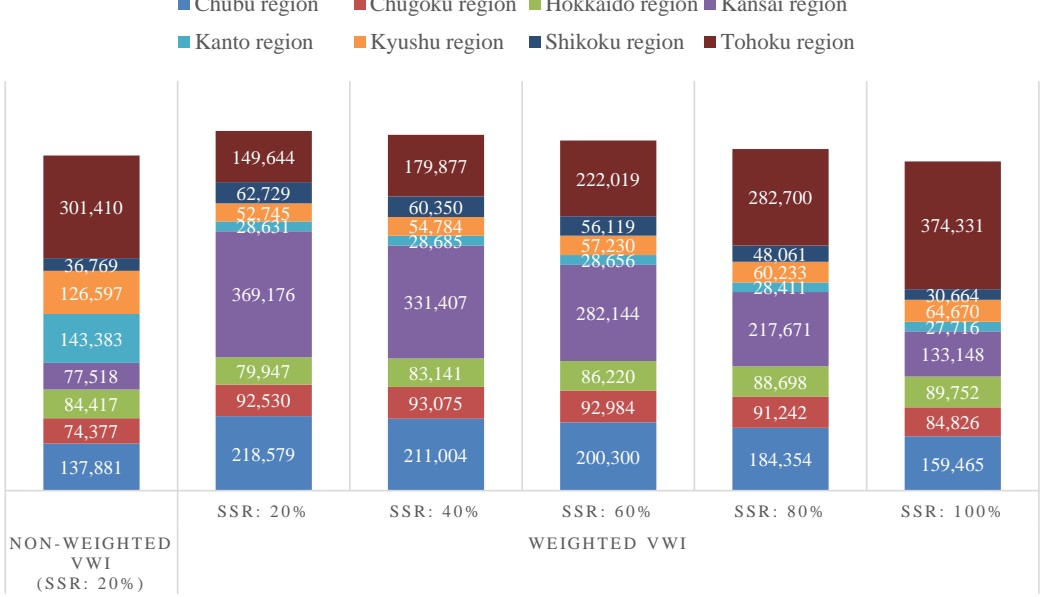

(b) Virtual water import in Osaka

**Figure 5** Annual virtual water import from various regions in Tokyo (a) and Osaka (b) during 2000–2012



## Appendix

**Figure A1** Non-weighted rice distribution among the 47 prefectures with a 20% self-supply scenario (thousand tons/yr)





**Figure A2** Weighted rice distribution among the 47 prefectures with a 20% self-supply scenario (thousand tons/yr)





**Figure A3** Weighted rice distribution among the 47 prefectures with a 60% self-supply scenario (thousand tons/yr)





**Figure A4** Weighted rice distribution among the 47 prefectures with a 100% self-supply scenario (thousand tons/yr)