# Peer review of "An assessment of domestic rice distribution for transboundary water-food management in Japan through virtual water trade"

_Hydrology and Earth System Sciences, 2019_

## Referee Comment (RC1) · Anonymous Referee #1 · 3 Sep 2019

This article describes an economic model that simulates the regional flows of rice on the Japanese market for this commodity. Rice flows are translated into virtual water flows using the water footprint concept. I do not recommend publication of this article in HESS for the following 3 main reasons:

1.) The main focus of the paper is the national rice market of Japan and its regional re-distribution flows, which are modelled using an economic approach. The only relation to water/hydrology is the water footprint calculation. I believe that the topic falls outside the core area that HESS usually covers.

2.) The paper has an almost exclusive focus on the Japanese market. I am not clear

about the value/impact of the paper for an international audience such as HESS's. The authors do not clearly expose what they see as the main contributions / generic insights that are applicable elsewhere.

3.) The authors do not clearly explain/expose what they see as methodological/technical innovation in their article. To my knowledge, neither the gravity model used to simulate rice flows nor the water footprint calculations are new contributions.

Apart from these main comments, I have a couple of technical concerns that should also be addressed, before this article can be considered for publication:

1.) The gravity model that is used to simulate regional rice flows in the Japanese market is not validated against any observational datasets. I strongly believe that the predictive skill of models has to be tested. This is a highly simplified approach and in order to establish confidence in the model results, validation is essential.

2.) It is unclear what the exact purpose of the water footprint calculations is apart from being able to show virtual water flows. The water footprint concept disregards the variability of water scarcity in space and time. The impact of abstracting virtual water from a water-scarce basin is larger than for a water-abundant basin. Impact is also more severe during periods of drought. The true water availability shadow price will be highly variable in space and time and this could be captured with a spatio-temporally resolved model of the Japanese water-food nexus.

---

## Referee Comment (RC2) · Anonymous Referee #2 · 13 Sep 2019

The article makes an assessment of the Japanese domestic distribution of rice and the virtual water trade that is linked to that distribution. Firstly the distribution of rice is estimated using the gravity model, after that the VWT is estimated by combining the distribution data with water withdrawals in the different prefectures. The study concludes that a lot of water is being transported (trough rice) from Japans northern regions to its southern regions and therefor it is important to consider the link between food and water in trans-boundary water management.

I'm not convinced that this methodology and conclusion justify publication in HESS, mainly because (1) the methodologies are not novel and (2) the study and its conclusions do not have an international scope.

Besides these issues, there are also some more technical issues I would like to point out. (1) The water withdrawals figures are provided by the local ministry. The distribution numbers however are completely based on the gravity model, without any form of validation. Therefor it remains a bit unclear whether or not the presented numbers are valid. Especially since the gravity model depends on an empirical parameter ($\alpha$), which the authors set to 2. Explaining this choice could help increase readers confidence. (2) Another issue is the absence of the influence of foreign exports to/from japan, since 10% of the domestic consumptions originates abroad. Discussing what the effect of this might be on the results would help.

- Technical Corrections:

Line 30: Please define self-sufficiency. I'm assuming it is something like "rice consumed/rice produced", but it would be good to specify that.

Line 53: Change "...also be to related water management..." to "...also be related to water management".

Paragraph starting on Line 53: break this paragraph into three shorter ones.

Line 109: Give a reference for this claim.

Line 130: Alpha is defined twice.

Line 139: Definition here of AE and RI is different from the ones on line 129.

Line 181: Table 1 should be Table 2.

Line 182: The number here does not add up to the numbers in table 2, should be 51.5.

---

## Author Comment (AC1) · 11 Oct 2019

Dear Editor and reviewers

We appreciate your comments and tried to apply all comments to the revised manuscript. The main comments seemed to be related to the validation of methodology and contribution of this study. First, we validated the methodology using observed data and added a new chapter "3.3 Validation of rice distribution simulation compared to 2016 observation". Second, we mentioned the application and contribution of this study in this chapter, for example, the importance of local distribution of food in the countries where produce large amount of domestic food products. However, there

were still limitations of validation in a scale issue and application of international trade issue, thus, we added more explanation of limitations in this study to the new chapter "4. Limitations, but possibilities". We also added more analysis of regional flows of rice and assessed the impacts of local food security on the regional dependency of rice in "3.2 Analysis of impacts of food security in prefectures on the entire rice distribution through various SSR scenarios" with additional Figure 2. To sum up, we added 1) Validation of methodology using regional data, 2) Limitations and contribution, and 3) Addition analysis of regional dependency of rice flow in revision. Please find the more details in revision notes in attached Supplement ZIP file.

All authors including myself have seen and approved this revised manuscript. I am looking forward to your response.

Thanks. Sincerely yours. __________________ Sang-Hyun Lee

Please also note the supplement to this comment:
https://www.hydrol-earth-syst-sci-discuss.net/hess-2019-284/hess-2019-284-AC1-supplement.zip